# A Skeleton-Line-Based Graph Convolutional Neural Network for Areal Settlements' Shape Classification

Yiyan Li [1,2,3], Xiaomin Lu [1,2,3,*], Haowen Yan [1,2,3], Wenning Wang [1,2,3] and Pengbo Li [1,2,3]

1   Faculty of Geomatics, Lanzhou Jiaotong University, Lanzhou 730070, China
2   National-Local Joint Engineering Research Center of Technologies and Applications for National Geographic State Monitoring, Lanzhou 730070, China
3   Gansu Provincial Engineering Laboratory for National Geographic State Monitoring, Lanzhou 730070, China
*   Correspondence: xiaominlu08@mail.lzjtu.cn

**Abstract:** Among the geographic elements, shape recognition and classification is one of the important elements of map cartographic generalization, and the shape classification of an areal settlement is an important part of geospatial vector data. However, there is currently no relatively simple and efficient way to achieve areal settlement classification. Therefore, we combined the skeleton line vector data of an areal settlement and the graph convolutional neural network to propose an areal settlement shape classification method that (1) extracts the skeleton line of the areal settlement to form a dual graph with nodes as edges, (2) extracts multiple features to obtain a graph representation of the shape, (3) extracts and aggregates the shape information represented by the areal settlement skeleton line using the graph convolutional neural network for multiple rounds to extract high-dimensional shape information, and (4) completes the shape classification of the high-dimensional shape information. The experiment used 240 samples, and the classification accuracy was 93.3%, with areal settlement shapes of E-, F-, and H-type achieving F-measures of 96.5%, 92.3%, and 100%, respectively. The result shows that the classification method of the areal settlement shape has high accuracy.

**Keywords:** areal settlement; skeleton lines; graph convolutional neural network; shape classification; cartographic generalization





## 1. Introduction

As a geospatial data type, an areal settlement plays an important role in the field of geographic information systems (GIS), usually appearing as a basic element in map updating [1,2], cartographic generalization [3–5], and so on. To improve the readability and quality of areal settlement data, it is necessary to classify the shape of the areal settlement.

As an important attribute of areal settlement elements, shape conveys a greater amount of information in the representation of geographical entities than color, texture, [6], etc. The shape recognition methods that national and international experts use for areal settlement can be broadly divided into two aspects:

1. Shape description and recognition of the two-dimensional surface area of an areal settlement, which is also divided into two parts. One is based on the description of the overall shape after gridding of the vectorial areal settlement. For example, the most straightforward way to describe the shape of a building is by constructing a description of the chain code [7], the shape context [8] descriptor, using the grid as the object. It also converts remotely sensed images into raster images, with wavelet descriptors based on raster data [9], or histogram statistics [10], both of which can provide a good description of image features. The other is the abstraction of areas within the areal settlement in special shapes instead. For example, the areal settlement is matched with a template in the shape of the alphabet [11,12]. This method is effective for the recognition of an areal settlement or the division of a two-dimensional plane into multiple triangular surface areas [13,14], a good way of describing global and local shape features. All of these methods partition the

areal settlement into regular geometric shapes, However, the results are not satisfactory for describing some of the more concave, convex, or complex areal settlements.

2. Description of the shape of the contour of the 2D-surface-shaped areal settlement boundary line, quantifying the contour with some special descriptive methods to describe the contour, for example, construction of a measure of the similarity of points on different contours through shape contexts [2], construction of the chordal eigenmatrix of a contour [15], multiscale shape representation of plane curves based on contour curvature [16], rotation functions [17], and description of the contours of building polygons using several spatial indicators [18]. These methods have the advantage of being computationally intuitive, but the two-dimensional surface contour information is too redundant and does not extract key concave and convex information for the classification of areal settlement shapes.

In recent years, deep learning has achieved excellent results in computer vision, natural language processing, and speech recognition [19–21]. Commonly used in the earth sciences to understand spatial processes and improve geographical insight, it is primarily combined with regular grid-like data to extract geometric features through convolutional neural networks. For example, convolutional neural networks combined with rasterized remote sensing images [22] are used to extract key information about important overpass structures or road intersections [23] or contour line data of one-dimensional trajectories are processed into regular raster-like data [24]. Both are prepared for the needs of neural networks for regular-like data.

However, the data for convolutional neural networks are supposed to be regularized grid-like data and cannot be applied directly to some irregularly shaped data. For this reason, some scholars have proposed graph convolutional neural networks (GCNs) [25], which use graph structures to construct irregular data, combined with convolutional neural networks. For example, graph data structures are constructed to recognize the shapes of river system clusters [26]. Constructing building group graph data helps to extract the geometric topological features among building groups and classify the internal regularity of building groups [27,28]. However, these methods have achieved good results in the distinction of groups of buildings rather than individual settlements. For large-scale individual areal settlements, based on theories such as the first law of geography and spatial association, the descriptive representation of elements in map space involves a local map structure consisting of neighboring points, neighboring edges, or arcs in a single geographic element [29]. Therefore, the combination of the local graph structure, the global perception of the overall structure, and the method of graph convolutional neural networks classifying the areal settlement by the shape of the alphabet [30]. These graph convolutional–neural–network-based areal settlement recognition algorithms can help accurately distinguish areal settlement shapes. However, the above method's data source is contour-based, which gives complete shape information on the shape of the areal settlement but has a redundant number of nodes compared to the skeleton lines [31]. Moreover, it is not very effective at distinguishing complex shapes of areal settlement. Inspired by these studies, this paper uses skeleton line data to better balance the characteristics of the overall alignment and local morphological features of a large-scale areal settlement. The graph structure is constructed using one-dimensional skeleton line data, and the low-dimensional shape information represented by the skeleton lines is aggregated and finally classified by a graph convolutional neural network through the extraction of skeleton line features. This solves the problem of classifying the shapes of complex-shaped large-scale areal settlements.

## 2. Methodology

### 2.1. Framework

In this paper, the data source is the areal settlement element. First, the skeleton line of the element is extracted and created as a graph data structure. Then, the graphs are annotated according to the shape categories of the residential elements. Next, the high-dimensional structural features of the residential elements are extracted by means of supervised graph learning using a graph convolutional neural network. Finally, a classifier

classifies the graph data, thus classifying the areal settlement shapes. This is divided into four processes, and the framework is shown in Figure 1.

(1) Data preprocessing: Areal settlement vector data were collected from the OSM dataset. Skeleton lines were constructed on the basis of the vector data of the areal settlement.
(2) Graph representation: A component dual-graph structure was constructed on the basis of the skeleton line graph structure.
(3) Feature extraction: Features of local and global structures representing the skeletal line dual graph of the areal settlement structure were extracted.
(4) GCN construction: The model architecture based on the GCN was designed. Then, the GCN was trained and tested, Finally, a fully connected layer as a classifier completed the Areal settlements' shape classification.

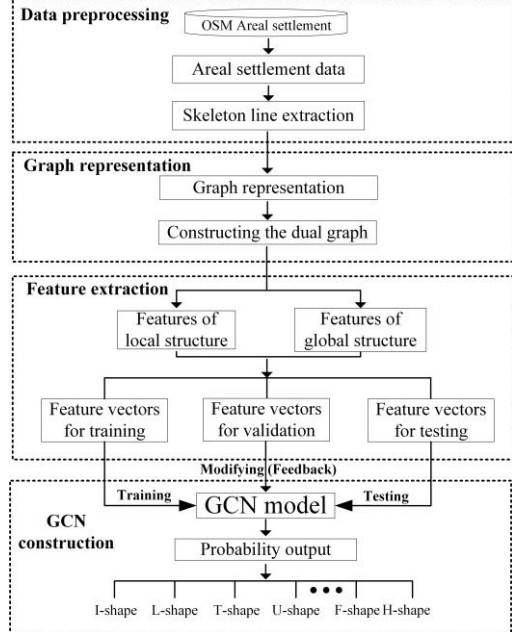

**Figure 1.** Framework.

## 2.2. Data Preprocessing

The data were obtained from the OpenStreetMap open-source dataset and divided into nine alphabetical shapes according to the Gestalt Principle. including E-type, F-type, I-type, and Y-type [32], as shown in Figure 2. Nine types of areal settlement were selected manually.

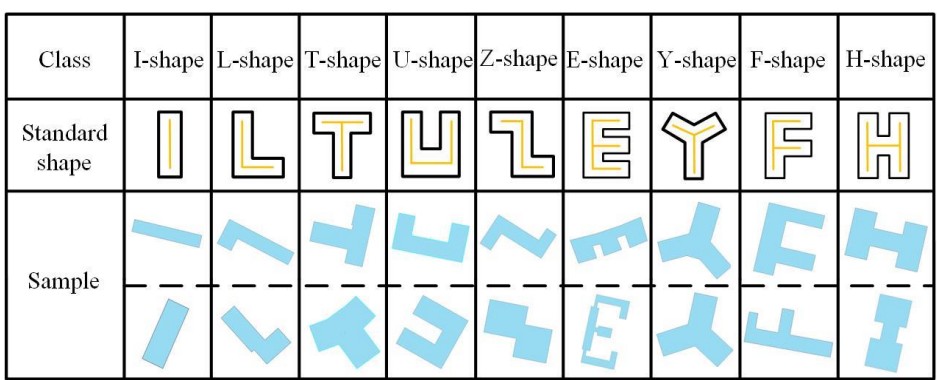

**Figure 2.** The nine shape classes of areal settlements.

The first step in shape classification using graph convolutional neural networks is to construct the areal settlement entities as graph data structures to create a graph repre-

sentation of the vector shape of the areal settlement. We use the skeleton line of the areal settlement entity to describe its shape. The Delaunay triangulation of the areal settlement is shown in Figure 3a, and the shape center of the Type III triangle (red triangular) is used as the end point of the skeleton line [33]. The end points are connected in sequence to form the skeleton line, as shown in Figure 3b.

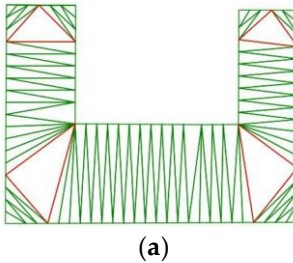 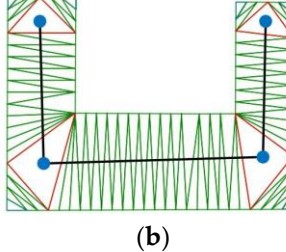

|       (a)       |       (b)       |

**Figure 3.** Extraction of areal settlement skeleton lines: (**a**) Delaunay triangulation; (**b**) The skeleton line of the areal settlement.

### 2.3. Graph Representation

The nodes of a graph neural network generally represent individuals in the network structure, e.g., people in a social network and functions in a software-structured network, and the edges represent the relationships between individuals [34]. Specifically for the areal settlement skeleton line structure, although its natural structure can be regarded as a graph structure, the relative location information within the areal settlement is embedded in the skeleton line and its connection relationship. Therefore, this paper takes the polygonal skeleton line segments of the areal settlement as the minimum shape unit, while using them as nodes of the graph, and the connection relationship between the line segments as edges to construct a graph data structure. As shown in Figure 4, the edge AB has a node D and the line segment DE is an edge.

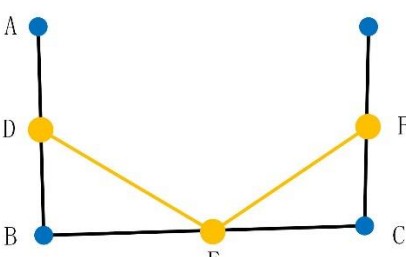

**Figure 4.** Dual graph of skeleton lines.

### 2.4. Feature Extraction

The node attribute feature extraction directly affects the shape representation of the whole graph. The skeletal lines of the residential area are used to construct the graph, mainly by describing the geometric features of the one-dimensional skeletal line combination and the topological relationships between the line segments, to extract their features of local and global structures and thus achieve an overall overview of the areal settlement.

#### 2.4.1. Local Features Extraction

Large-scale areal settlements are simple two-dimensional surface shapes. The essential local information is the length and the direction of the extension. Therefore, we extract the skeleton line length and azimuth as local feature. The length of the skeleton line segment AB is $L_1$; the azimuth of the skeleton line segment is $\alpha$ as shown in Figure 5.

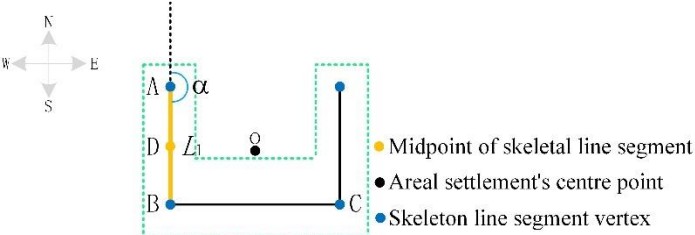

**Figure 5.** Local features extraction of node D.

The length of the line segment consisting of the azimuth angle $\alpha$ denotes the direction of the line segment AB in the two-dimensional plane. In this study. the angle from the northern end of the line due north, turned clockwise to the target line, is noted as the azimuth of the line and takes values in the range 0.360°.

The shape features must be normalized as they should remain constant under translation, rotation, and reduction. The length of the skeleton line segment $L_1$ is normalized by dividing it by the sum of the total lengths of the skeleton lines $D$. The azimuth angle $\alpha$ is normalized by dividing it by 360° according to the range of values of the azimuth angle $\alpha$, as follows:

$$\overline{L_1} = \frac{L_1}{D} \tag{1}$$

$$\overline{\alpha} = \frac{\alpha}{360} \tag{2}$$

### 2.4.2. Global Features Extraction

Since the skeleton lines represent the main and convex features of the areal settlement, the features between adjacent skeleton lines in residential areas cannot be ignored and offer great advantages in the classification of complex shapes. As shown in Figure 6a, the midpoint D of the boundary line segment AB is connected to the center point of the areal settlement. The global features are extracted as follows: The length of the line OD is $L_2$. The turning angle of the line segment OD to OE is $\beta$. It represents the angle from OD to OE, which is positive in the counterclockwise direction and negative otherwise. Area S1 is denoted by the line connecting the center point of the areal settlement to the points on either side of the skeleton line segment (Figure 6b).

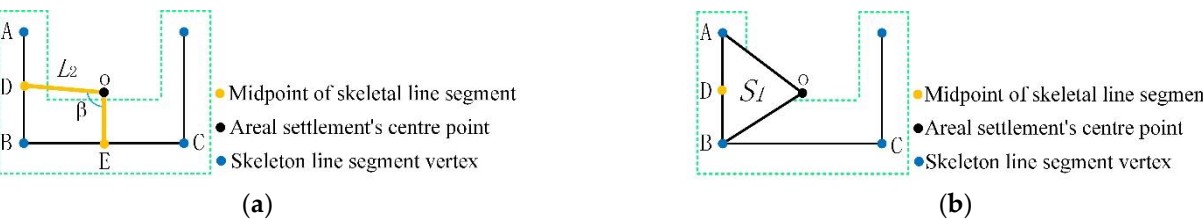

(**a**)                                                                         (**b**)

**Figure 6.** Global features extraction of node D: (**a**) length and turning angle; (**b**) local area.

The global features need to be normalized. The length $L_2$ is normalized by dividing the sum of $L_{\text{sum}}$ the angle $\beta$ is normalized by dividing 360°, and the area $S_1$ by dividing the sum of the local area $S_{\text{sum}}$, as follows:

$$\overline{L_2} = \frac{L_2}{L_{sum}} \tag{3}$$

$$\overline{\beta} = \frac{\beta}{360} \tag{4}$$

$$\overline{S} = \frac{S_1}{S_{sum}} \tag{5}$$

### 3. Structure of the GCN-Based Residential Classification Model

After gaining the feature vectors for the training set, the validation set, and the test set, the data are inserted into a graph convolutional neural network. When constructing an areal settlement into a graph structure, the skeleton lines of the areal settlement need to be mapped to the graph structure. In this research, the skeleton line characteristics of the surface settlement morphology are embedded in the nodes of the graph and the main morphology of the areal settlement is expressed by the graph structure of the skeleton line. The attributes of the nodes in the graph carry local and overall morphological features of the surface settlement, which are extracted by different methods because the residential areas have different shapes. Thus, when performing convolutional operations on the graph, features of the shape can be extracted from different aspects, and in classification prediction, the shape features are aggregated by a fully connected neural network and their category is predicted according to the obtained probability values of the shape.

Figure 7 displays the GCN architecture used in this study. In the forward propagation process, the initial input signal is the feature matrix $X_0 \in R^{N \times D}$ of the skeleton line, where $N$ is the number of skeleton line nodes and $D$ is the number of skeleton line features. Deep information about the areal settlement is extracted through graph convolution and activation calculation of hidden layers. The number of neurons in each hidden layer is I, that is, the number of convolution kernels. Finally, the probability $Y \in R^{1 \times 9}$ of the nine skeleton lines is output in the $(L+1)^{th}$ layer, or the fully connected layer. The graph convolution and activation calculation formula of the GCN hidden layer is as follows:

$$H_j^{[l+1]} = ReLU\left(\sum_{i=1}^{I}\left(\hat{D}^{-\frac{1}{2}}\,\hat{A}\,\hat{D}^{-\frac{1}{2}}\,H_i^{[l]}W^{[l+1]}\right) + b_j^{[l+1]}\right) \tag{6}$$

where $ReLU(\cdot)$ represents a rectified linear function, $H_i^{[l]}$ is the output of the $i^{th}$ neuron in the $l^{th}$ layer, and $H_j^{[l+1]}$ is the output of the $j^{th}$ neuron in the $(L+1)^{th}$ layer. $H^{[0]}$ is set as $X_0$. $W^{[l+1]} \in R^{I_l \times I_{l+1}}$ and $b_j^{[l+1]} \in R^{I \times I_{l+1}}$ are the trainable weight matrix and bias, respectively, which are the training parameters of the GCN. Furthermore, $\hat{A} = I_N + A$ is the adjacency matrix and $\hat{D}$ is the degree matrix of $\hat{A}$.

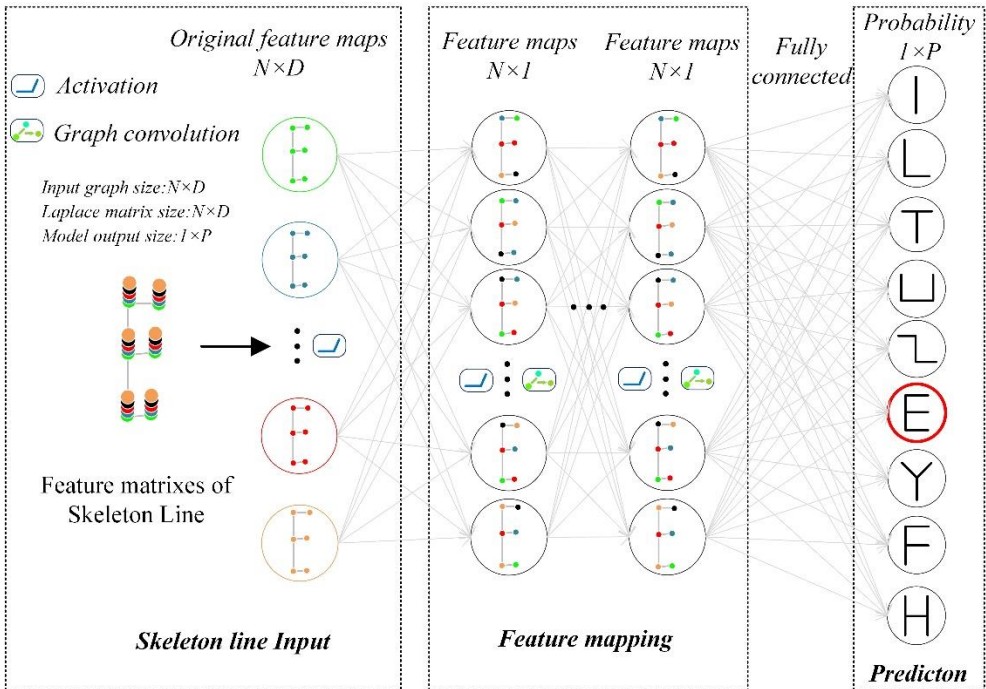

**Figure 7.** Architecture of the GCN.

After the graph convolution operator is applied to $L$ hidden layers, the output of the $(L+1)^{th}$ layer is $H^{[L+1]} \in R^{1 \times 9}$, or Y. The $(L+1)^{th}$ layer is the fully connected layer, which is calculated as follows:

$$Y = Softmax\left(reduce\_mean\left(H^{[L]}W^{[L+1]} + b^{[L+1]}\right)\right) \tag{7}$$

where $W^{[L+1]} \in R^{I_L \times 9}$ and $b^{[L+1]} \in R^{1 \times 9}$ represent the learning weights and bias, respectively, of the $(L+1)^{th}$ layer and $\left(H^{[L]}W^{[L+1]} + b^{[L+1]}\right) \in R^{N \times 9}$ calculates the probability of a different classification at each node of the graph. The final classification is judged by a global probability value. Therefore, the $reduc\_mean(\cdot)$ function is used to calculate the mean along the first dimension (node dimension). Then, the normalized exponential function $Softmax(\cdot)$ is used to obtain the output probability Y of the nine classifications. In backpropagation, to maximize the efficiency of the model, the cross-entropy function is used to calculate the model loss, and the Adam algorithm is selected for the optimization.

## 4. Experiments and Analysis

### 4.1. Experimental Environment and Data

The experiments were implemented in Python 3.7 and Pytorch 1.6. The platform used was Microsoft Win10 64-bit operating system, CPU Intel(R) Core(TM) i7-3740QM at 2.70 GHz, 8 GB of RAM, and 500 GB of hard disk.

In this experiment, there are 200 of each areal settlement type and a total of 1800 samples of 9 shapes are selected. Firstly, 120 samples are randomly selected from each of the 9 types, i.e., a total of 1080 samples are used as the training set; 40 areal settlement data are randomly selected from each category in the remaining 720 samples; a total of 360 samples are used as the validation set; and the remaining 360 samples are used as the test set. Due to the limitations of the number of datasets and the effect of a more accurate response model, the training set, the validation set, and the test set are divided into a ratio of 6:2:2.

In this experiment, the structure of the constructed skeleton line graph is labeled according to the shape of the areal settlement and a dataset with labels is obtained. For feature extraction of the nodes in the graph, the attributes of the corresponding nodes are extracted according to the method described in Section 2.4, and the extracted attributes are assigned to the corresponding nodes to complete the pre-processing of the graph data.

### 4.2. Model Structure and Parameter Settings

This experiment uses a 5-layer model with 4 convolutional layers and 1 linear classification layer. Model parameters are shown in Figure 8. The graph is constructed using the method proposed in this paper for constructing residential graph data and extracting node attributes. The constructed graph with labels is input into the model in batches, the weight parameters are initialized, the model parameters are learned, and the weight parameters are updated by back propagation, with a maximum of 350 iterations. After experimentation, the model with a learning rate of 0.005 and a training batch of 10 is selected. To prevent overfitting, training is stopped when the number of successive iterations is 100 and the accuracy of the validation set is no longer increasing.

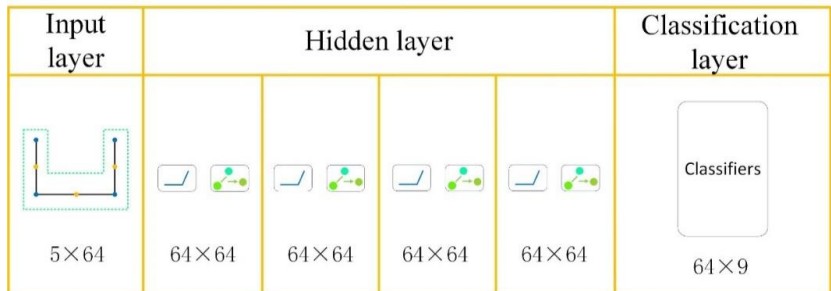

**Figure 8.** Parameters of the areal settlement classification model.

In the first graph input layer, the input feature dimension is 5 and the embedding vector dimension is 64. After the graph convolution operation, the output value of the first layer is passed to the next layer and the ReLU function is used as the activation function. Layers 2 to 4 of the graph convolutional layers are used as the hidden layers of the graph convolutional neural network. The input and output vectors of the model have a dimension of 64, and the activation function is a ReLU function. The fifth layer is a linear classification layer, which uses a fully connected neural network and a softmax function as the classification layer to classify the output of the hidden layer through the classification layer. The output is the predicted shape classes of the areal settlement, corresponding to the shape classes in the dataset.

Figure 9 presents the validation accuracy and loss of the training and validation sets in this experiment. From the results, it can be concluded that the loss value decreased to 0.069 at the end of the training and the accuracy value of the validation set was 95.3%, while the model was able to converge well after 250 iterations. From the accuracy value of 95.3%, it can be concluded that the model has good generalization ability, has good sensitivity to the skeleton line of the face areal settlement, and can effectively classify the shape of the areal settlement.

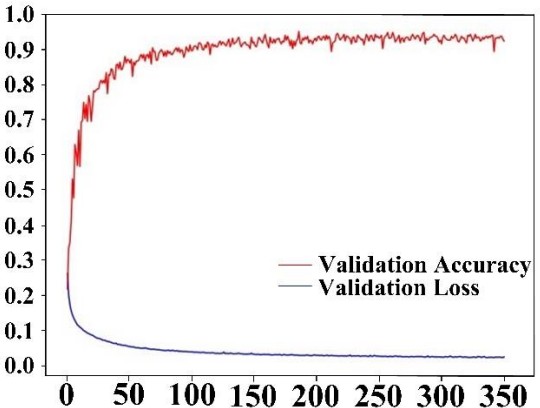

**Figure 9.** Changing curve of loss and validation accuracy in GCN model training.

### 4.3. Sensitivity Analysis of Model Parameters

In the training of neural networks, the training batch size has an impact on the performance of the model. The decrease in each gradient is determined by the overall sample data for each training batch. In the experiments, the accuracy of the model training results was further studied by varying the training batch size from 10 to 500, as shown in Figure 10, from which it can be concluded that the accuracy of the model training results did not decrease significantly when the training batch size was increased from 10 to 500. However, the accuracy was the highest when the training batch size was 10, and there was a significant convergence after 300 epoch.

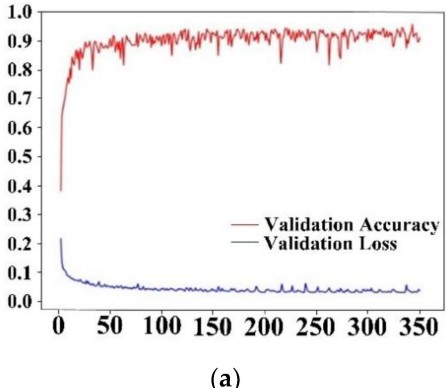

(**a**)

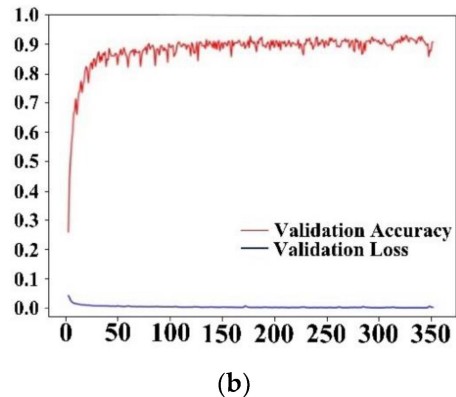

(**b**)

**Figure 10.** *Cont.*

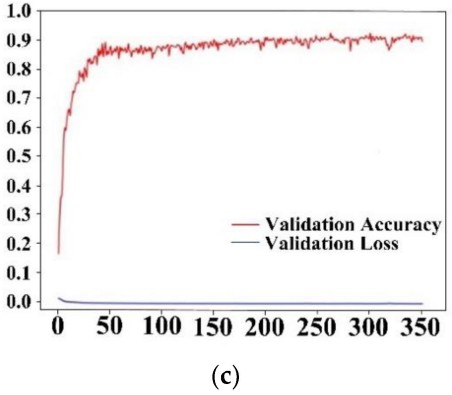

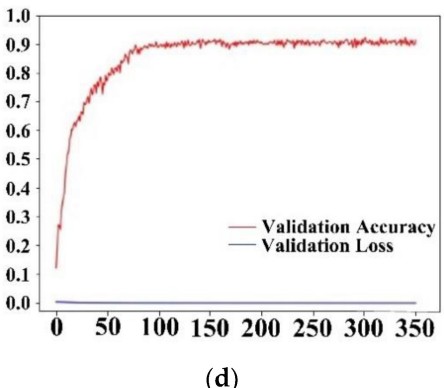

(**c**) (**d**)

**Figure 10.** The validation accuracy of the model with training batches of different sizes; (**a**) batch 10, accuracy 0.9361, (**b**) batch 50, accuracy 0.9278, (**c**) batch 100, accuracy, 0.9222, and (**d**) batch 500, accuracy 0.9028.

With a learning rate of 0.005 and a training batch size of 10, keeping other conditions constant, this experiment investigated the effect of the number of layers of the model and the dimensionality of the shape-embedding vector on the performance of the model. From Figure 11, it can be concluded that the accuracy of the model classification could be improved by increasing the depth of the model and the dimension of the shape-embedding vector under certain conditions, and it can be seen from the figure that the highest accuracy can be obtained by choosing a model depth of 4 and an embedding vector dimension of 64 as parameters.

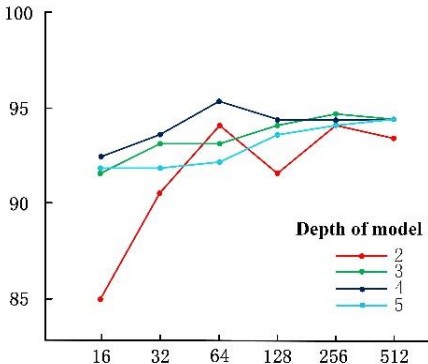

**Figure 11.** Test of the number of hidden layers and vector dimension.

After the experimental study to derive the hyperparameters when the model is optimal, the effect of model depth on model performance is further investigated. The optimal hyperparameters of the model were identified when the other hyperparameters of the model were certain, such as a training batch of 10, an embedding vector dimension of 64, a learning rate of 0.005, and a maximum of 350 iterations.

*4.4. Classification Accuracy Analysis*

To test the effectiveness of the areal settlement shape classification model, it is necessary to give a test set, use the model to predict each sample in the test set, and calculate an evaluation score on the basis of the results of the model's predicted classification. For classification problems, common evaluation criteria include accuracy, precision, recall, and F-measure values [26]. This paper uses accuracy to evaluate how well the model classifies all classes as a whole, and for each class, the performance estimates are evaluated using precision and recall. The accuracy of a class is the proportion of all samples predicted to be in that class that are predicted correctly. The recall of a class is the proportion of all samples truthfully labeled as being in that class that are predicted correctly. The result

is shown in Table 1. The F-measure is the summed mean of the accuracy and recall and represents a metric given when both are equally important.

$$\text{Accuracy rate}: \ \text{P} = \frac{\textit{Number of Correct Classifications}}{\textit{Number of classifications in the Model}} \tag{8}$$

$$\text{Recall rate}: \ \text{R} = \frac{\textit{Number of correct classifications}}{\textit{Artificial number of classifications}} \tag{9}$$

$$\text{Measurement values}: \ \text{F} = \frac{2 \times P \times R}{P + R} \tag{10}$$

**Table 1.** The results of the areal settlement classification test.

| Class | Artificial Number of Classifications | Number of Classifications in the Model | Number of Correct Classifications | *P*/(%) | *R*/(%) | F |
|---|---|---|---|---|---|---|
| I-shape | 40 | 40 | 40 | 100 | 100 | 100 |
| L-shape | 40 | 40 | 37 | 92.5 | 92.5 | 92.5 |
| T-shape | 40 | 44 | 39 | 88.6 | 97.5 | 92.8 |
| U-shape | 40 | 39 | 37 | 94.9 | 92.5 | 93.7 |
| Z-shape | 40 | 39 | 39 | 100 | 97.5 | 98.7 |
| E-shape | 40 | 43 | 38 | 88.4 | 95 | 91.6 |
| Y-shape | 40 | 40 | 39 | 97.5 | 97.5 | 97.5 |
| F-shape | 40 | 36 | 35 | 97.2 | 87.5 | 92.1 |
| H-shape | 40 | 39 | 39 | 97.5 | 97.5 | 97.5 |
| Total | 360 | 360 | 343 | 95.3 | 95.3 | 95.3 |

*4.5. Comparison to Other Methods*

To compare the differences between the skeleton-line-based graph convolutional neural network method and other similar methods, this paper conducted comparative experiments using the contour-line-based graph convolutional neural network method [29] and the turning angle function method.

On using the contour-line-based graphical convolutional neural network method, four features are extracted from the nodes: 1.the length of contour line AB is $L_1$, 2.the azimuth of contoured line segments is $\alpha$ (Figure 12a), 3.the length of the line segment OP is $L_2$, and 4.the angle of the turn of the line OP to OQ is $\beta$ (Figure 12b). The input feature dimension of the model is 128 × 4. A four-layer hidden layer structure with 128 hidden layers is used to finally obtain 128 × 9 dimensional embedding vectors, and the training results of the model are shown in Figure 13.

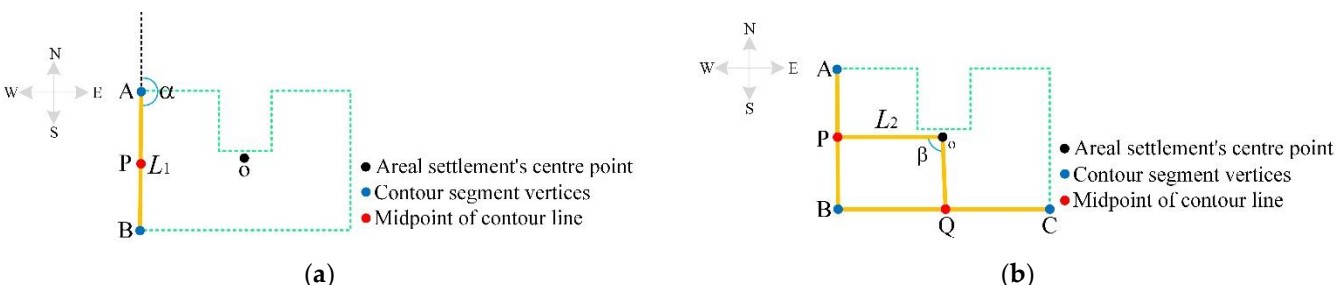

(a)  (b)

**Figure 12.** Feature extraction; (**a**) local feature extraction of node P; (**b**) global feature extraction of node P.

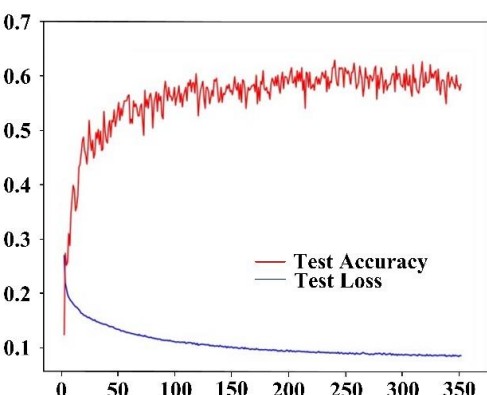

**Figure 13.** Changing curve of loss and validation accuracy in GCN model training.

In the comparison experiment of the turning angle function, the principle is shown in Equation (11). For polygon *A*, the tangent angle of any point O on its boundary along the reference direction (e.g., *x*-axis) is v. The turning angle function $f(s)$ means the change in the tangent angle along the boundary of *A* is inverse or counterclockwise relative to the arc length *s*. A total of $120 \times 9 = 1080$ experimental data in 9 classes were subjected to accuracy calculations, yielding a matching accuracy of 0.438.

$$L_P(A, B) = ||f_A(s) - f_B(s)||_P = \left( \int_0^1 |f_A(s) - f_B(s)|^P ds \right)^{\frac{1}{P}} \tag{11}$$

A comparison of the experimental results of the skeleton-line-based graph convolutional neural network, the contour-line-based graph convolutional neural network, and the turning angle function is shown in Table 2.

**Table 2.** Comparison of methods.

| Methods | GCN Based on Skeleton Lines | GCN Based on Contour Lines | Turning Function |
|---|---|---|---|
| Accuracy rate | 95.3% | 64.4% | 47.8% |

The main reason for this discrepancy is that the skeleton line accurately portrays the overall orientation and convexity of the settlement, whereas the contour line cannot accurately portray the convexity of the settlement. The features of the skeleton line achieve an apparent classification effect after dimensional expansion and multilayer convolution of the graph convolutional neural network. In contrast, the classification of the graph convolutional neural network is not affected by spatial factors such as the starting point and the starting angle of the areal settlement, so the accuracy of the graph convolutional neural network based on skeleton line is higher than that of the turning angle function.

*4.6. Applications*

To verify the usability of the model, the skeleton-line-based shape classification model for residential maps in this study was applied to identify the shape of some areal settlements in Lanzhou City, with experimental data from 1:5000 Lanzhou City building data. In the vector graph of the areal settlement in Lanzhou City, this paper selected some areal settlements as the test objects, such as the yellow part in Figure 12. The test sample dataset contained 240 samples, the selected areal settlement was manually annotated, and the trained model was used to classify this dataset. The classification result accuracy reaches about 93.3%, the accuracy rate is 93.3%, and the recall rate is 93.3%. The model classification results are shown in Table 3. As can be seen from Figure 14, in practice, buildings are mostly rectangular in shape, of which Type F and Type E are well classified and can all be recognized.

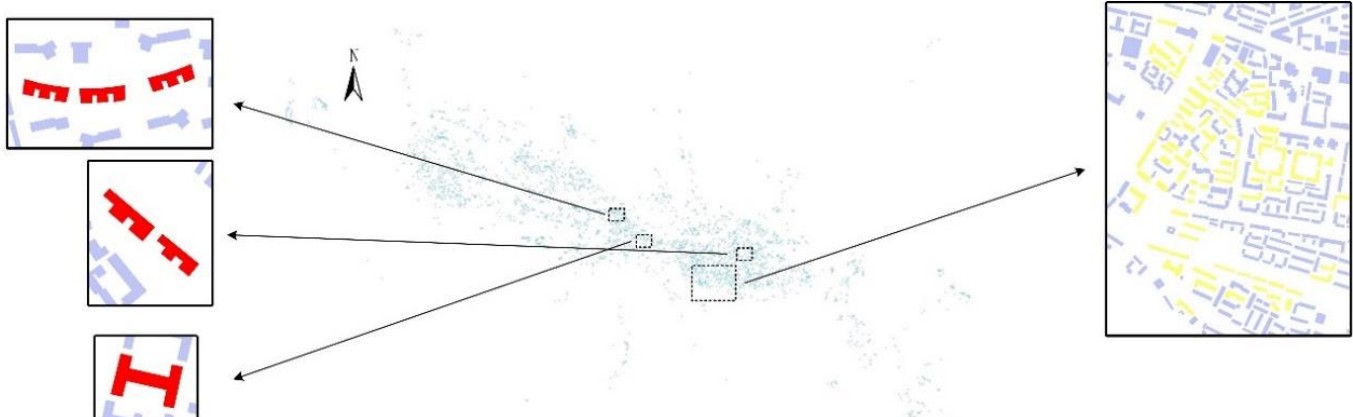

**Figure 14.** Experimental data of Lanzhou City.

**Table 3.** Classified statistics of some settlements in Lanzhou.

| Class | Artificial Number of Classifications | Number of Classifications in the Model | Number of Correct Classifications | P/(%) | R/(%) | F |
|---|---|---|---|---|---|---|
| I-shape | 88 | 84 | 84 | 100 | 95.5 | 97.7 |
| L-shape | 40 | 42 | 40 | 95.2 | 100 | 97.5 |
| T-shape | 17 | 12 | 11 | 91.7 | 64.7 | 75.9 |
| U-shape | 38 | 38 | 36 | 94.7 | 94.7 | 94.7 |
| Z-shape | 9 | 14 | 8 | 57.1 | 88.9 | 69.5 |
| E-shape | 15 | 14 | 14 | 100 | 93.3 | 96.5 |
| Y-shape | 6 | 9 | 5 | 55.6 | 83.3 | 66.7 |
| F-shape | 13 | 13 | 12 | 92.3 | 92.3 | 92.3 |
| H-shape | 14 | 14 | 14 | 100 | 100 | 100 |
| total | 240 | 240 | 224 | 93.3 | 93.3 | 93.3 |

On combining the data from the training phase in the previous section, the classification results of the model in the experimental set are slightly lower than those in the test set, but the accuracy is still strong. In distinguishing complex shapes, such as E, F, and H shapes, the measured values are mainly concentrated between 95% and 100%. The main reason is that the skeleton line is a better representation of the main body and convexity of the settlement than the areal settlements, thus providing a more accurate description of the local features of a complex-shaped settlement. However, for the T- and Y-shaped areal settlements, with similar skeleton line structures, the measured values are 75.9% and 66.7%, respectively. In the next step of the study, the training set and the experimental dataset will be further increased in order to reflect the accuracy of the model classification more objectively.

## 5. Conclusions

Areal settlements' shape classification is a classical and challenging issue in large-scale topographic maps. This study proposed a graph convolutional neural network shape classification method based on the skeleton lines of the areal settlement. The method uses a skeleton line of areal settlement elements as the basis for constructing the calculation graph. The geometric features of the skeleton line segments are extracted as the nodal attribute features of the computational graph, using the skeleton line segments as the smallest unit to express the shape features of the faceted areal settlement in a graph structure. To achieve shape classification of an areal settlement, a graph with labels and node attributes is used as input for the GCN classification mode, and the graph convolutional neural network is used to discriminate between high-dimensional shape features through supervised training. Lastly, a fully connected classifier can accurately classify the areal settlement's shape.

The experimental results show that the method has a high discriminative ability for areal settlement shapes.

The method uses areal settlement skeleton lines as input data to construct an end-to-end classification model. Validation with real data sets has demonstrated that the skeleton line can effectively represent both local and global features and has a high discriminative ability for the shape of alphabetical areal settlement. The distinction of complex shapes of areal settlement such as F- and E- and H-types is particularly effective and outperforms other existing methods. However, the objects of the areal settlement in this method are large-scale. Shape variation in small- and medium-scale areal settlements is one of the main areas to be improved and is proposed to be studied in depth in combination with the method of graph convolutional neural networks.

**Author Contributions:** Y.L. and X.L. conceived the skeleton-line and Graph Convolutional Neural Network together; Y.L. collected the data, designed the experiments, and wrote the manuscript; W.W. and P.L. offered significant contribution to result evaluation; H.Y. modified the manuscript. All authors have read and agreed to the published version of the manuscript.

**Funding:** This research was funded by the National Natural Science Foundation of China (No. 41930101, 42161066), the Industrial Support and the Program Project of Universities in Gansu Province (grant no.2022CYZC-30).

**Institutional Review Board Statement:** Not applicable.

**Informed Consent Statement:** Not applicable.

**Data Availability Statement:** Not applicable.

**Conflicts of Interest:** The authors declare no conflict of interest.

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
