# Peer review of "A Skeleton-Line-Based Graph Convolutional Neural Network for Areal Settlements’ Shape Classification"

_applsci, doi:10.3390/app121910001_

Round 1

Reviewer 1 Report

Introduction

  • Page 1: Indicate the meaning of the acronym GIS and include a brief definition of the concept

  • Page 2: In order to have a better point of comparison with these works, mention more exactly the efficiency obtained by the works of references [26], [27], [28] and [29].

Methodology

  • Framework: Mention the 5 features extracted and justify the choice of that number of features used in this work

  • Framework: Mention the name of the classifier used.

  • Data preprocessing: What was the criteria for selecting those nine types of faceted areal settlements?

  • Graph representation: If examples of what nodes and edges represent in a graph neural network are mentioned, you must include references to works where the aforementioned is demonstrated

Structure of the GCN-Based residential Classification Model

  • Justify the choice of some neural network parameters such as the cross-entropy function used to calculate the model loss and the Adam algorithm selected for the optimization.

Experiments and Analysis

  • Experimental Environment and Data: Mention the justification for the use of the Python programming language for this work

  • Experimental Environment and Data: Is the 6:2:2 ratio recommended for the dataset for this type of work or how was that choice made?

  • Classification Accuracy Analysis: It is mentioned that common evaluation criteria include accuracy, precision, recall, and F-measure values. Reference should be made to works that use these indicators.

Conclusions

  • The conclusion section should be completely restructured. Currently it only mentions a summary of what has been done in this work. In this section, specific conclusions generated from the results obtained in the work must be presented.

Reviewer 2 Report

To recognize and classify areal settlement shapes, the authors have proposed a graph convolutional neural network shape classification method based on the skeleton lines of the areal settlement. The results of the article are quite sound and interesting, however, the paper is in need of some improvements suggested below:

 1.      The abstract and conclusion are too short and do not summarize the present study in a systematic fashion. Moreover, the motivation of the study should be made more lucid.

2. The authors have not provided any motivation in the introduction section. They have not clarified why they consider this problem and what are the advantages and key features of the proposed technique. Moreover, an updated and complete literature review should be conducted and should appear as part of the Introduction.

3.   The results and findings should be compared to and discussed in the context of relevant work available in the open literature.

Reviewer 3 Report

The research is well handled and presented. I would like a minor correction in the attached file.
